# Investigation of Cerebral Autoregulation Using Time-Frequency Transformations

**DOI:** 10.3390/biomedicines10123057

**Published:** 2022-11-28

**Authors:** Vladimir Semenyutin, Valery Antonov, Galina Malykhina, Vyacheslav Salnikov

**Affiliations:** 1Almazov National Medical Research Center, Ministry of Health of Russia, Polenov Neurosurgical Research Institute, 12 Mayakovsky Street, Saint-Petersburg 191014, Russia; 2Department of Higher Mathematics, Peter the Great St. Petersburg Polytechnic University, Saint-Petersburg 195251, Russia; 3Higher School of Cyber-Physical Systems and Control, Institute of Computer Science and Control, Peter the Great St. Petersburg Polytechnic University, Saint-Petersburg 195251, Russia

**Keywords:** cerebral autoregulation, transcranial doppler, Mayer waves, wavelet transform, wavelet coherence, phase shift

## Abstract

The authors carried out the study of the state of systemic and cerebral hemodynamics in normal conditions and in various neurosurgical pathologies using modern signal processing methods. The results characterize the condition for the mechanisms of cerebral circulation Institute of Computer Science and Control, Higher School of Cyber-Physical Systems and Control regulation, which allows for finding a solution to fundamental and specific clinical problems for the effective treatment of patients with various pathologies. The proposed method is based on the continuous wavelet transform of systemic arterial pressure and blood flow velocity signals in the middle cerebral artery recorded by non-invasive methods of photoplethysmography and transcranial doppler ultrasonography. The study of these signals in real-time in the frequency range of Mayer waves makes it possible to determine the cerebral autoregulation state in certain diseases before and after surgical interventions. The proposed method uses a cross-wavelet spectrum, which helps obtain wavelet coherence and a phase shift between the wavelet coefficients of systemic arterial pressure signals and blood flow velocity in the Mayer wave range. The obtained results enable comparing the proposed method with that based on the short-time Fourier transform. The comparison showed that the proposed method has higher sensitivity to changes in cerebral autoregulation and better localization of changes in time and frequency.

## 1. Introduction

The future development of personalized medicine is determined by the synergy of knowledge and efforts of scientists from several fields of medicine, physiology, mathematics, and computer science. Approaches based on signal processing, computer modeling, and machine learning complement the main instrumental methods for studying biological processes and allow a deeper understanding of the mechanisms of human disease and personalized treatment strategies. Despite the increased popularity of artificial intelligence, machine learning, and signal processing approaches, a lot of effort still needs to be put into preparing them for clinical implementation. The use of modern methods of signal processing and neural networks allows for increasing the possibilities of non-invasive methods to measure and control regulation processes. Features of integrating general and personalized heterogeneous data is a complex task, the solution of which should comply with accepted ethical and legal standards.

Cerebral autoregulation (CA) maintains relatively constant cerebral blood flow despite perfusion pressure change. The study of the CA phenomenon was difficult when using invasive methods for assessing changes in cerebral blood flow and perfusion pressure, which are not applicable in clinical practice [1]. The introduction of transcranial Doppler ultrasound into practice and the development of non-invasive methods for assessing systemic blood pressure (BP) and dynamic CA made it possible to conduct studies directly at the patient’s bedside [2,3,4,5].

CA functions through myogenic, metabolic, and neurogenic mechanisms. Usually, there is a distinction between static and dynamic CA. Static CA characterizes changes in cerebral blood flow with long-term changes in perfusion pressure, whereas dynamic CA is associated with relatively rapid fluctuations in BP. The CA system dampens these fluctuations, which manifests itself in the presence of consistency and phase shift between fluctuations in BP and blood flow velocity (BFV) in the arteries at the base of the brain. Early diagnosis of CA violations and their res allows for the prevention of a number of ischemic and hemorrhagic complications.

The rapid development of digitalization in medicine, the use of computers, and modern software packages make it possible to develop and apply methods of mathematical statistics, digital signal processing, and machine learning in medical research to study the processes of CA [6,7,8,9,10].

The spectral analysis of BFV and BP showed the presence of four relatively stable fluctuations: heart rate (0.65–1.4 Hz), respiratory excursions (0.15–0.65 Hz), Mayer systemic waves (0.08–0.12 Hz), and intracranial B-waves (less than 0.05 Hz) [11,12]. The slow arterial pressure oscillations originally described by S. Mayer [13] had in anesthetized rabbits a frequency of 6 to 9 cycles/min, i.e., 0.1 to 0.15 Hz, which is slower than the frequency of spontaneous sympathetically mediated arterial pressure oscillations in conscious rabbits (~0.3 Hz). The term Mayer waves are widely used now.

For continuous assessment of the CA state in real-time, some authors use several methods of digital signal processing [14,15,16]. They select the M-waves of the BFV and BP signals using an ideal filter and calculate the coherence value, the phase shift (PS) in the coherence intervals for the received signals. The violation of CA leads to an increase in the bandwidth of the autoregulatory filter in the range of M-waves and, as a result, to an increase in coherence and a decrease in the PS between the BFV and BP oscillations. This protective mechanism plays a significant role in the functioning of the brain. The disadvantage of the real-time implementation of this method is the insufficiently good localization of the coherence intervals.

Other authors apply the method of assessing the CA by calculating the cross-correlation coefficient between the BFV and BP signals in the M-wave range. In-phase slow fluctuations of these signals with low efficiency of the CA lead to an increase in the transmission of M-waves, whereas the cross-correlation between BFV and BP is close to unity. While in the normal state of the CA, the cross-correlation decreases. The disadvantage of this method is the dependence of the CA estimation results on the shape of the cross-correlation function and on the phase shift angle between BFV and BP, which can vary from 0.8 to 1.4 radians, affecting the cross-correlation coefficient [17,18,19].

A number of publications show the possibility of using the wavelet transform of the BFV and BP signals to estimate the coherence and phase shift in the wavelet decomposition spaces that correspond to the M-wave range [20,21].

Further development of this approach seems promising. Therefore, the goal of our study is to develop an algorithm for diagnosing disorders of the regulation of cerebral circulation, using the functions of wavelet coherence and phase shift of the BFV and BP signals in the wavelet decomposition spaces corresponding to the range of M-waves, and to implement this algorithm in the measuring information system in real-time.

The paper aims to develop a method for determining the current state of the patient’s cerebrovascular autoregulation system in real-time, including “at the patient’s bedside”. Real-time CA monitoring, along with conventional monitoring, seems to be a promising method for improving individualized patient care. Early diagnosis of impaired cerebral autoregulation and its restoration allows the prevention of a number of ischemic and hemorrhagic complications.

Evidently, this method reflects our knowledge of the autoregulation process, having higher sensitivity and lower delay. The method based on the cross-correlation function, which is being developed by scientists [22,23], indirectly determines the state of autoregulation, but it is not sensitive enough. The result of this method depends on the form of the cross-correlation function. Transcranial Doppler sonography (TCD) is the main non-invasive method for the continuous recording of cerebral blood flow, which makes it possible to assess the rate of CA in real-time with a simultaneous non-invasive recording of BP [24]. TCD is implemented as a portable, bedside, non-invasive diagnostic tool used to assess cerebral hemodynamics in real-time [25].

Assessment of CA using NIRS-only methodology seems feasible in critically ill sedated/coma patients after incorporating methodological improvements. As the authors of the article [26] state, the NIRS-only methodology has the advantage that it is non-invasive and does not require monitoring of arterial blood pressure. However, the method has not yet entered into practice and requires further validation. R. Panerai et al. [27] use an approach to measuring CA based on transfer function analysis. They showed that although the TCD is only capable of measuring blood flow index and not true blood flow, the measurement results are representative and widely used to assess dynamic CA [28,29].

A new assessment of dynamic CA using magnetic resonance imaging technology makes it possible to evaluate both global and spatially differentiated values of the autoregulation index [27]. Big data in intensive care units expand the opportunities for neurocritical care and helps prevent secondary brain damage [30].

## 2. Materials and Methods

### 2.1. Patients and Healthy Volunteers

We examined 6 patients aged 41 to 63 with arteriovenous malformations of the brain, 6 patients aged 52–72 with stenosis of the brachiocephalic arteries, and 9 healthy volunteers, including four women and five men, aged 19 to 35 from students and staff of the Almazov National Medical Research Center (ANMC), Saint Petersburg, Russia, who did not have any cardiovascular, pulmonary, and cerebrovascular pathology in accordance with the Protocol of 5 April 2010, approved by the Ethics Committee of the ANMC and Carnet-consensus. At the same time, we performed non-invasive monitoring of BP using digital photoplethysmography (CNAP, Graz, Austria) and BFV in both middle cerebral arteries using transcranial doppler ultrasonography (MultiDop X, DWL, Singen, Germany,) in the supine position under the control of end-tidal CO_2_.

We performed hypercapnic (breathing with a 5% mixture of CO_2_ with air for two minutes) and hypocapnic (hyperventilation, providing a significant decrease in BFV, which characterizes an increase in the tone of the distal arteries and arterioles) tests. The tests were used to evaluate changes in the state of the CA.

All studies were approved by the Ethics Committee of ANMC.

### 2.2. Time-Frequency Analysis of Signals Characterizing CA

The numerical method for assessing the CA state is determined by the presence of consistency between the fluctuations of BP and BFV in the range of M-waves. In our survey, we used the range of M-waves, in which fluctuations are primary, and the phase shift between BP and BFV characterizes the state of CA. The analysis of other frequency ranges is not the subject of this article. The state of the target audience should be diagnosed both offline and online. When working in real-time, the signal analysis is performed within a frame sliding along the signals. In both cases, it is required to isolate the coherent components of the signal in the given frequency range to determine the coherence coefficient of the signals and the phase shift between them. The offline method allows us to analyze the signal after the end of the measurements.

#### 2.2.1. Short-Time Fourier Transform

The traditional mathematical method that allows exploring the signal in the frequency domain, which is based on the classical Fourier transform, implicitly assumes that the signals are stationary in time. For non-stationary signals, the windowed Fourier transform is usually used, which in the case of the Gaussian window is called the Gabor transform [31].

The short-time Fourier transform (STFT) is performed within the frame sliding along the BP and BFV signals, which we denote as $x\left(n\right),y\left(n\right),n=1,\ldots ,N_{frame}$, where $N_{frame}$ is the frame length. Signals x(n) and y(n) are mixed with other physiological signals and noise. Therefore, when calculating the signal coherence characteristics and the phase angle, the signals are smoothed within the frame. For smoothing, the frame is divided into L windows $x\left(n\right),y\left(n\right),n=1,\ldots ,N_{win}$, of length $N_{win}$, and the signal characteristics averaged over all windows within each frame are calculated. The Hann window is used for processing. The window is shifted within the frame by the amount equal to half the length of the window $N_{shift}=\frac{1}{2}N_{win}$.

For the centered signals in the windows, the $N_{win}$–point Fourier transform X(k), Y(k)*,* is calculated, where k are discrete frequencies. The mutual spectral density $S_{x,y}\left(k\right)=X^{*}\left(k\right)Y\left(k\right)/N_{win}$, its modulus $\left|S_{x,y}\left(k\right)\right|$, and the phase $\Theta_{x,y}$ are related by: $S_{x,y}\left(k\right)=\left|S_{x,y}\left(k\right)\right|exp\left(-j\Theta_{x,y}\right)$.

The spectral densities of signals x(n) and y(n) are the following:(1)$$S_{x,x}\left(k\right)=\frac{X\left(k\right)X^{*}\left(k\right)}{N_{win}};\;\;$$
(2)$$S_{y,y}\left(k\right)=\frac{Y\left(k\right)Y^{*}\left(k\right)}{N_{win}}.\;$$

The calculations are repeated for each shifted position of the window; as a result, we obtain *k* values of the mutual and intrinsic spectral densities. The frame is characterized by the smoothed averaged spectral densities obtained for each window within the frame:(3)$${\hat{S}}_{x,x}\left(k\right)=\frac{1}{L}\overset{L}{\underset{l=1}{\sum }}S_{x,x}\left(k\right),{\hat{S}}_{y,y}\left(k\right)=\frac{1}{L}\overset{L}{\underset{l=1}{\sum }}S_{y,y}\left(k\right),{\hat{S}}_{x,y}\left(k\right)=\frac{1}{L}\overset{L}{\underset{l=1}{\sum }}S_{x,y}\left(k\right),\;$$

The coherence function is calculated from the smoothed spectral densities for each frame using the formula:(4)$$\gamma_{x,y}\left(k\right)=\sqrt{\frac{{\left|{\hat{S}}_{x,y}\left(k\right)\right|}^{2}}{{\hat{S}}_{x,x}\left(k\right){\hat{S}}_{y,y}\left(k\right)}}$$

The average phase shift ${\hat{\Theta }}_{x,y}$ between the signals is obtained from the relation ${\hat{S}}_{x,y}\left(k\right)=\left|{\hat{S}}_{x,y}\left(k\right)\right|exp\left(-j{\hat{\Theta }}_{x,y}\right).$

The signals are represented as a mixture with other physiological signals and with noise. Therefore, when calculating their spectra, averaging is proposed to suppress the noise. For this purpose, the data frame is divided into windows, and the Fourier spectrum is calculated for each window. The resulting spectra are averaged within the frame boundaries. The offset of a window within the frame depends on the type of window. The frame offset is set equal to the window offset. This allows us to speed up the calculations.

After calculating the coherence function and the phase shift of the signals for all L windows of the current frame, the frame is shifted by the number of samples Nshift=Nwin2, then the number of windows in every frame equals to L=NframeNshift.

The coherence function and the phase shift for each next frame are obtained from the corresponding values of the previous frame by removing the characteristics of the first window and adding the characteristics of the last window within the new frame. Then for each m-th frame, the coherence function $\gamma_{x,y}\left(k,m\right)$ and the phase shift function ${\hat{\Theta }}_{x,y}\left(k,m\right)$ are obtained, where k is the frequency sample number and m is the sample number by time. The discreteness of time is counted by Nshift=Nwin2. Therefore, the result of the preprocessing algorithm is a two-dimensional time-frequency function of the signal coherence $\gamma_{x,y}\left(k,m\right)$ and a two-dimensional time-frequency function of the phase shift ${\hat{\Theta }}_{x,y}\left(k,m\right)$.

When analyzing the state of cerebral autoregulation in real-time, time resolution, which is determined by the $N_{shift}$ frame shift in time, becomes important. Frequency resolution matters when fine-tuning Mayer’s frequency to determine the signal coherence. The STFT-based method has a constant resolution in time and frequency over the entire frequency range of the BP and BFV signals, which does not always ensure the best analysis results. The wavelet transform of the signal makes it possible to detect localized discontinuous periodicity associated with certain disorders of the CA [32].

#### 2.2.2. Continuous Wavelet Transform

The disadvantage of the windowed Fourier transform is the fact that the window size is chosen once and remains constant; it does not adjust to the spectral properties of the signal. To eliminate this shortcoming, we have replaced the traditional STFT-based approach with the wavelet transform-based approach. A continuous wavelet can be interpreted as a set of harmonic functions with a window that changes its size depending on the frequency band of the signal under study.

Continuous Wavelet Transform (CWT), using a harmonic wavelet basis, is better applied for detecting the harmonic components of the BP and BFV signals [33]. Wavelets of this type include the complex Morlet wavelet, the complex Pole wavelet, and the real wavelet representing the difference between Gaussians.

The idea behind the CWT is to use the wavelet as a band-pass filter. The CWT of the signal x(t) is defined as the convolution with the scaled and normalized wavelet.
(5)$$\hat{X}\left(a,b\right)=\frac{1}{\sqrt{a}}\int_{-\infty }^{\infty }x\left(t\right)\psi^{*}\left(\frac{t-b}{a}\right)dt;$$
where $\psi^{*}\left(.\right)\;$denotes the complex conjugation, $\frac{1}{\sqrt{a}}\psi \left(\frac{t-b}{a}\right)$ is the normalized wavelet function whose parameter b corresponds to the time shift, and the parameter a>0 specifies the scaling.

After replacing the integral with the sum for discrete calculations n=tδt, the relation for the CWT coefficients becomes:(6)$$\hat{X}\left(s,n\right)=\overset{N-1}{\underset{n^{\prime }=0}{\sum }}x\left(n\right)\psi^{*}\left(\frac{\left(n-n^{\prime }\right)\delta t}{s}\right)$$

By the convolution theorem, the wavelet coefficients can be calculated more efficiently as the inverse discrete Fourier transform (DFT) of the product of the Fourier transforms of the signal and the wavelet coefficients in accordance with the formula:(7)$$\hat{X}\left(s,n\right)=x\left(n\right)*\psi \left(n\right)=F^{-1}\left[F\left(x\left(n\right)\right)\cdot F\left(\psi \left(n,s\right)\right)\right]$$
where $F^{-1}\;$is the operator of the inverse discrete Fourier transform and s is the scale number.

To analyze the BP and BFV signals, we choose the Morlet wavelet, which is used in medicine for cardiogram and encephalogram analyses more often than other wavelet bases since it ensures the rapid determination of changes in non-stationary signals.

Continuous non-orthogonal wavelets are effective for the analysis of time series and aperiodic shifts: the Morlet complex wavelet, the Pole complex wavelet, and the real Gaussian difference wavelet. The Morlet wavelet is determined by the formula:(8)$$\psi \left(t\right)=\pi^{-\frac{1}{4}}e^{-i\omega_{0}t}e^{-\frac{t^{2}}{2}}$$
with parameter $\omega_{0}=6$. The Morlet wavelet in the frequency domain has the form $\hat{\psi }\left(s\omega \right)=\pi^{-\frac{1}{4}}e^{-\frac{{\left(s\omega -\omega_{0}\right)}^{2}}{2}}.$

### 2.3. Wavelet Transform of Signals Characterizing CA

The state of the CA is characterized by a phase shift between the BP and BFV separately for the right and left hemispheres of the brain in the presence of signal coherence in the decomposition space corresponding to the M-wave range, where the coherence is maximum. When describing the algorithm, we use the designation $\xi_{n}\;$for the sequence of readings of the BP signal and $\zeta_{n}$ for the BFV.

The algorithm for calculating the coherence and phase shift of signals by CWT, which is applied to two discrete centered signals $x_{n}=\xi_{n}-\frac{1}{N}\overset{N-1}{\underset{n=0}{\sum }}\xi_{n}$ and $y_{n}=\zeta_{n}-\frac{1}{N}\overset{N-1}{\underset{n=0}{\sum }}\zeta_{n}\;$, is measured with the use of optical sensors. The calculation is applied to data frames of length: $=N=2^{floor\left(\log_{2}\left(N^{\prime }\right)\right)+1}$, where $N^{\prime }$ is the initial length of the signal frame to be analyzed.

For each signal frame $x_{n}$ and $y_{n}$, we calculated the DFT:(9)$${\hat{x}}_{k}=\sum_{n=0}^{N-1}x_{n}e^{-i2\pi kn},\;{\hat{y}}_{k}=\sum_{n=0}^{N-1}y_{n}e^{-i2\pi kn}$$
where k=0,1,2…N−1 denotes the frequency index.

The wavelet decomposition parameters, such as the minimum scale $s_{0}$, the maximum decomposition level J, specific decomposition levels j=0.1…J and the scale vector $s_{j}\;$are determined by the formulas: $s_{0}=2\delta t$, $J=\delta j^{-1}\log_{2}(\frac{Nx\delta t}{s_{0}}),$ $s_{j}=s_{0}2^{j\delta j}.$

The localization of wavelets in time and frequency makes it possible to associate the pseudo-frequencies $f_{j}$ with the scale $s_{j}$: $f_{j}=\frac{\omega_{0}}{2\pi s_{j}}\;$with allowance for the Fourier factor $\eta_{0}=\frac{\omega_{0}}{2\pi }$.

The circular frequency at each expansion scale is determined by the formula:(10)$$\omega_{k}=\{\begin{array}{c} \frac{2\pi k}{N\delta t},\;\;k\leq \frac{N}{2}\\ -\frac{2\pi k}{N\delta t},\;\;k>\frac{N}{2} \end{array}$$

The parameter $\omega_{0}$ affects the resolution in time and frequency. The temporal resolution decreases as the frequency resolution increases. To analyze the signals, the Morlet wavelet parameter ω_0_ = 6 is chosen, which approximately corresponds to the respiratory rate.

DFT of the analytical Morlet wavelet defined by the formula:(11)$$\hat{\psi }\left(s\omega_{k}\right)=\pi^{-\frac{1}{4}}e^{-\frac{{\left(s\omega -\omega_{0}\right)}^{2}}{2}}H\left(\omega \right),$$
contains the Heaviside function $H\left(\omega_{k}\right)=\{\begin{array}{c} 1,\;\omega_{k}>0\\ 0,\;\omega_{k}\leq 0 \end{array}$. On each expansion scale, the wavelet coefficients are normalized according to the formula:(12)$$\hat{\psi }\left(s\omega_{k}\right)=\sqrt{\frac{2\pi s}{\delta t}}{\hat{\psi }}_{0}\left(s\omega_{k}\right).$$

Element-by-element multiplication of the Fourier image of the analyzed signal and the Fourier images of wavelets at each decomposition level and the subsequent inverse DFT transformation allows us to obtain the coefficients of the wavelet decomposition of the signals $x_{n}$ and $y_{n}$ in the following form:(13)$$c_{x}\left(n,s\right)=\overset{N-1}{\underset{n=0}{\sum }}{\hat{x}}_{k}{\hat{\psi }}^{*}\left(s\omega_{k}\right)e^{i\omega_{k}n\delta t},$$
(14)$$c_{y}\left(n,s\right)=\overset{N-1}{\underset{n=0}{\sum }}{\hat{y}}_{k}{\hat{\psi }}^{*}\left(s\omega_{k}\right)e^{i\omega_{k}n\delta t}.$$

The calculation of the wavelet coefficients of the Fourier domain enables reducing the complexity of the calculations.

#### 2.3.1. Smoothing

The signals x(n) and y(n) are mixed with other physiological signals and noise; it is necessary to perform smoothing when calculating the wavelet transform of the signals.

Time smoothing is conveniently performed in the frequency domain using the Gaussian window. To do this, on each scale, $s=1:\;N_{s}$, we multiply the Fourier transform of the wavelet coefficients $F\left(c_{x}\left(n,s\right)\right),\;s=1:\;N_{s}$ by the filter impulse response $h\left(n,s\right)=e^{-\frac{1}{4}s^{2}\omega {\left(s\right)}^{2}}$ and perform the inverse Fourier transform of the results:(15)$${\tilde{c}}_{x}\left(n,s\right)=F^{-1}\left[h\left(n,s\right)\cdot F\left(c_{x}\left(n,s\right)\right)\right],$$
where $F^{-1}\;$is the inverse DFT operator, ${\tilde{c}}_{w}\left(n,s\right)$ are the smoothed wavelet coefficients.

Scale smoothing was performed by averaging for a rectangular window:(16)$${\tilde{\tilde{c}}}_{x}\left(n,s\right)=\overset{L_{s}-1}{\underset{l=0}{\sum }}{\tilde{c}}_{x}\left(n,s+l\right)\frac{1}{L_{s}},\;$$
(17)$${\tilde{\tilde{c}}}_{y}\left(n,s\right)=\overset{L_{s}-1}{\underset{l=0}{\sum }}{\tilde{c}}_{y}\left(n,s+l\right)\frac{1}{L_{s}}.\;$$
where $L_{s}$ is the length of the window.

Consequently, we obtained the wavelet coefficients of two signals x(n) and y(n) smoothed in time and scales ${\tilde{\tilde{c}}}_{x}\left(n,s\right)\;и\;{\tilde{\tilde{c}}}_{y}\left(n,s\right)$.

The distribution of the wavelet coefficients in the area of Mayer waves is more peaked than the Gaussian distribution. This fact should be taken into account, when smoothing the coefficients in order to obtain more accurate results of the coherence analysis.

#### 2.3.2. Cross-Wavelet Transform

The wavelet cross-spectrum characterizes the total energy of two signals, which is non-zero if the two signals correlate with each other and disappear if the two signals are independent:(18)$${\tilde{\tilde{c}}}_{x,y}\left(n,s\right)={\tilde{\tilde{c}}}_{x}^{*}\left(n,s\right){\tilde{\tilde{c}}}_{y}\left(n,s\right).\;$$

In the general case, the wavelet cross-spectrum is a complex function, which is represented as an amplitude and a phase:(19)$${\tilde{\tilde{c}}}_{x,y}\left(n,s\right)=\lfloor {\tilde{\tilde{c}}}_{x,y}\left(n,s\right)\rfloor exp\left(arctg\frac{Im\left({\tilde{\tilde{c}}}_{x,y}\left(n,s\right)\right)}{Re\left({\tilde{\tilde{c}}}_{x,y}\left(n,s\right)\right)}\right).$$

The coherence or consistency of two signals can be defined as the modulus of the normalized cross-spectrum. Coherence defines a linear relationship between two signals. The value of coherence varies from zero to one. The square of the normalized coherence value is determined by the formula:(20)$$H_{x,y}^{2}\left(n,s\right)=\frac{{\left\lfloor {\tilde{\tilde{c}}}_{x}^{*}\left(n,s\right){\tilde{\tilde{c}}}_{y}\left(n,s\right)\right\rfloor }^{2}}{{\left\lfloor {\tilde{\tilde{c}}}_{x}^{*}\left(n,s\right)\right\rfloor }^{2}{\left\lfloor {\tilde{\tilde{c}}}_{y}\left(n,s\right)\right\rfloor }^{2}}.\;$$

In the presence of the signal coherence specified by the condition $H_{x,y}^{2}\left(n,s\right)\geq 0.6$, the local phase shift of the signals can be obtained by the following formula:(21)$$\theta_{x,y}\left(n,s\right)=arctg\frac{Im\left({\tilde{\tilde{c}}}_{x,y}^{*}\left(n,s\right)\right)}{Re\left({\tilde{\tilde{c}}}_{x,y}\left(n,s\right)\right)}.$$

The coefficient, 0.6, corresponds to the scale decorrelation length for the Morlet wavelet [33,34]. A. Kulaichev [35] empirically showed in the analysis of encephalograms that the value of the coherence coefficient should be greater than 0.6.

The value of the phase shift allows us to define the delay between two coherent signals. Statistical data processing and calculations were carried out using the Matlab computer program. The significance of differences in values was assessed using the Student’s *t*-test.

The sensitivity of phase shift changes during the tests was determined by the formula: $\eta =\frac{\left|\theta_{ext-}\bar{\theta }\right|}{\bar{\theta }}$, where $\theta_{ext-}$ is the extreme value of PS when it affects on autoregulation, $\bar{\theta }$ is the average value of PS when there is no effect on autoregulation.

## 3. Results

### 3.1. Results of the Wavelet Analysis of the BP and BFV Signals

BP and BFV measurements were obtained synchronously with a time interval of 0.01 s after analog-to-digital conversion. The received BP and BFV formed into frames were processed using the continuous wavelet transform. For each frame, CWT was performed using the Morlet wavelets. As a result, the coherence values (20) were obtained for the sequence of frames. The phase shifts (21) were calculated for those samples for which the coherence values met the condition $H_{x,y}^{2}\left(n,s\right)\geq 0.6$.

The results obtained for one frame are shown in Figure 1 The arrows show the phase angles between the BP and BFV signals in the left middle cerebral artery against the background of the magnitude-squared coherence between these signals. The magnitude-squared coherence value is shown in color according to the color bar. High coherence is observed in the area of pseudo-frequencies that correspond to the heart rate, respiratory rate, and M-waves. It is typical for a healthy volunteer (Figure 1a) to obtain a stable value of the coherence and shear angle. For a patient with an arteriovenous malformation in the basin of the left middle cerebral artery (Figure 1b), the shift angle was significantly smaller (0.5 rad). These plots were obtained using the Matlab coherence function wcoherence().

Figure 2 shows the values of the average phase angle between the BP and BFV obtained during the observation time (6 min) for both study groups. For each subject, the average values of the phase shift angle on both sides and the confidence limits at the level of two standard deviations ($\theta_{x,y}\pm 3\sigma$) are shown. Healthy volunteers are characterized by more stable and symmetrical values of the angles. In patients with arteriovenous malformation (AVM) in the region of the left middle cerebral artery, the values of the shear angles are almost half of that on the right, and their values are more variable, with increased standard deviation and wider confidence intervals.

### 3.2. System of Neuro Care Monitoring

The system of Neuro Care Monitoring (NCM), designed to assess the state of the CA in real-time, implements the algorithm presented here for the frame-by-frame determination of the phase angle between the BP and BFV in the Mayer wave range in areas of coherence.

Figure 3 shows a block diagram of the NCM system applying transducers of the multichannel system (1) Multi Dop X (DWL, Singen, Germany) for continuous non-invasive assessment of BFV in the arteries of the brain base with the use of transcranial Doppler ultrasonography and the Instrument (2) CNAP (Austria, Graz), for measuring systemic blood pressure by photo-plethysmography. The measurement results after the analog-to-digital conversion and the formation of frames in Data Input Device (3) are fed to the input of the Computing Unit (4), which implements the developed algorithm in real-time, sequentially processing data frames using the wavelet transform. Parameter Input Interface (5) allows us to set a number of algorithm parameters: frame size and offset, coherence threshold value, frequency from the Mayer range, etc. To document the results of the study, you can enter the date and time of the study, patient or volunteer data, preliminary diagnosis, etc. The results of patients’ examinations can be displayed (6), printed as a document (7), saved in a data file, and sent to the database (8). The software interface is designed to set calculation parameters and view results in real time.

The standalone application (software) was written in the C programming language to acquire a reasonable calculation result in real-time. To get a better user-friendly interface, we used the National Instruments Labwindows/CVI software developing environment [https://ni.com/cvi]. (accessed on 20 August 2022). Its run-time library has a wide range of control and representation features we utilized in our application. Moreover, it has a set of data analysis functions that we also use in our software. The screenshots in Figures 5 and 6 were taken by our application.

The sensors were always placed standardly bilaterally for registration of BFV in the initial segments of the middle cerebral arteries. The frame size for STFT was 214 = 16,384 samples with an interval of 0.01 s, and the frame duration was 163.84 s. Inside the frame, the sliding Hamming window was used, the size of which was 1024 samples, and the offset was 512. Accordingly, the number of windows inside the frame was 32.

The frame size for the CWT transform was 214 = 16,384 samples with an interval of 0.01 s, and the frame duration was 163.84 s. The duration of the study was 35 min. The frequency range of M-waves approximately corresponds to the frequency range of 0.08–0.12 Hz. The study of signals in the range of M-waves shows that a more accurate frequency setting within the M-range allows for increased sensitivity of the algorithm. The higher frequency resolution of the wavelet transform makes it possible to determine the phase shift in a specific space with maximum signal coherence.

Figure 4 shows the displayed results of examining a healthy volunteer during standardized exercise. The hypercapnic test was based on breathing for 2 min with a carbogen—5% mixture of CO_2_ with air, and the hypocapnic test was based on rapid, deep breathing for 1 min, leading to a significant decrease in CO_2_ in exhaled air.

The graphs results of measuring the BP and BFV in the left middle cerebral artery (MCA) and the right MCA are shown in red, blue, and dark green, respectively. The scale of these parameters is shown on the left. Graphs of the phase shift between the BP and the left BFV and between the BP and the right BFV, calculated on the basis of wavelets, are shown by the yellow and red lines, respectively. These plots can be compared with similar plots derived from the FFT, as shown by the blue and green lines. Curves on the graphs show that the CWT-based algorithm is more sensitive to PS changes during hypercapnic (1) and hypocapnic (2) trials, and it has better time localization. The arrow pointing up indicates the beginning of the hypercapnic test (1), while the arrow pointing down indicates its end. Similarly directed arrows indicate the beginning and end of the hypocapnic test (2).

Sensitivity analysis of phase shift changes showed that the CWT-based method is more sensitive. The sensitivity to the hypercapnic test using STFT for the left and right hemispheres averaged 0.065 and 0.060, and for the CWT method—0.11 and 0.10, respectively. The sensitivity to the hypocapnic test of the STFT method averaged 0.33 and 0.31, and the sensitivity of the CWT method was 0.46 and 0.47, respectively.

For smaller samples of examined persons, we determined the reliability of the difference between the methods based on CWT and STFT using the t-test.

As a result of testing, we obtained the following:-The hypercapnic test led to a greater relative decrease in PS on both sides for the CWT method than for the STFT method; the magnitude of the decrease was 14.6 ± 6.6% for CWT and 8.2 ± 4.5% on the left (*p* = 0.022), and for STFT—14.4 ± 5.8% and 8.2 ± 4.2% on the right (*p* = 0.014).-The hypocapnic test led to a greater relative increase in PS on both sides for the CWT method than for the STFT method; the magnitude of the increase was 44.4 ± 22.7% for CWT and 28.8 ± 17.3% for STFT on the left (*p* = 0.035), 45.9 ± 24.8% and 28.2 ± 17.9% on the right (*p* = 0.041).

Figure 5 shows the results of the survey of a patient with a unilateral CA disorder in the region of the stenotic artery, detected in real-time during the hypercapnic test (1) at time intervals indicated by the number (2). The arrow pointing up indicates the beginning of the hypercapnic test (1), while the arrow pointing down indicates its end. Similarly directed arrows indicate the beginning and end of the hypocapnic test (2). Figure 4 and Figure 5 adopted the same designation as the dependency graphs.

Hypercapnic and hypocapnic tests are not always attractive for patients with pathology. Therefore, such tests were carried out for one patient. The sensitivity of the CWT-based method was also slightly better than that of the STFT method.

To validate the CWT-based method, we obtained BP and BFV measurements from volunteers and patients within 8–10 min. These data were used to evaluate wavelet coherence and PS for 11 healthy volunteers, six patients with malformation, four patients with stenosis, and 10 patients with thrombosis (six patients before surgery). Reliable determination of the left or right affected part of the artery was a criterion for the quality of the algorithm. The PS between the BP and BFV signals in the wavelet domain of the Mayer waveband for AVM patients is shown in Figure 2b.

Figure 6 presents the average values of PS for patients with carotid stenosis.

The analysis of the results of the PS assessment for patients with stenosis, shown in Figure 6, allows us to conclude that Patient 1 has normal CA, Patients 2 and 6 have a very low PS value, indicating a violation of CA, Patients 2 and 3 have stenosis of the left, and in Patients 4 and 5—stenosis of the right internal carotid artery. The asymmetry of CA for Patients 2–5 is determined with a reliability greater than 0.999.

Tests on healthy volunteers and patients were performed with a data frame length of 16,384 samples, obtained in 2.7 min. The number of scales of the wavelet transform is 16 with a 12 voices number. Taking into account the Fourier factor 0.9549, it turned out that in the Mayer wavelength range, we had nine frequencies f = [0.124, 0.117, 0.110, 0.104, 0.098, 0.093, 0.088, 0.083, 0.078] Hz. This number of frequency gradations allowed us to fine-tune the algorithm. The frequency resolution varied along the M-wave range. At the beginning of the range at the frequency of 0.12 Hz, it was 0.007 Hz, and at the end of the range at the frequency of 0.08 Hz, it was 0.005 Hz. The time resolution was 0.01.

The algorithm based on the short-time Fourier cross-spectrum had the worst frequency resolution, which was the same over the entire range, equal to 0.04 Hz for our example.

## 4. Discussion

When monitoring CA in humans under various pathological conditions or in normal conditions, it is advisable to use non-invasive assessment methods based on retrospective cross-spectral and cross-correlation analysis of slow fluctuations in systemic and cerebral hemodynamics. When conducting functional tests and diagnosing emergency conditions in intensive care, it is necessary to apply a method that would provide prompt information about the state of the CA in real time, with the best resolutions in time and frequency. In our previous study [36,37], we proposed an algorithm based on the short-time cross-Fourier spectrum and the coherence spectrum, which makes it possible to obtain the estimated characteristics with constant scale resolution in time and frequency. The results received under standardized loads on the cerebral circulatory systems showed the possibility of assessing the state of cerebral autoregulation in real-time, and it first established the advantages of the wavelet analysis to collect reliable data on the phase shift between M-waves of BFV and BP.

The present study is different, as it allows monitoring with both higher sensitivity and better resolution in the time-frequency domain due to the use of continuous wavelet transform of the signals. In order to prove this, the authors carried out special tests that affect CA.

At certain time slots, the condition $H_{x,y}^{2}\left(n,s\right)<0.6$ may not be satisfied, there is no coherence, and the phase shift cannot be calculated. In such cases, the graphs (Figure 5 and Figure 6) may have gaps, the number of which should be reduced. To do this, we propose the use of a more accurate Mayer frequency fit. In our case, the wavelet transform using 12 voices allows us to search at 14 pseudo-frequencies in the Mayer waveband. The subrange selection criterion may be the largest proportion of values suitable for calculating the phase shift. When scanning subranges of Mayer waves, it turned out that the relative proportion of coherence intervals varies from 0.35 to 0.82 for the BP and BFV in the left hemisphere and from 0.50 to 0.88 for the BP and BFV in the right hemisphere. Thus, increasing the resolution in frequency improved the result of the study.

The proposed method was implemented as an algorithm for the operation of the analytical NCM. Studies of healthy volunteers and patients with CA disorders carried out with the NCM indicated the reliability of the results of a non-invasive assessment of the CA rate in real time. By simultaneously monitoring indicators of systemic and cerebral hemodynamics and CA rate, the developed analytical measuring system allowed in real-time enhancing non-invasiveness, the efficiency of an objective assessment of a person’s state in the norm, and the identification of a group of patients with CA disorders and a high risk of complications. All this can determine tactics and continuous monitoring of treatment results.

Accumulation and centralization of the digitized research data of many patients and the combination and formation of the knowledge base in the field of medicine and physiology can serve as a prerequisite for further research. The observations related to one patient contribute to a deeper understanding of the disease mechanisms and personalization of treatment. The integration of general and personalized data can make it possible to generate big data for the analysis of hidden patterns, where it is possible to use machine learning methods.

One of the trends for the development of methods for studying CA characteristics is to supplement the traditional methods of spectral, correlation, and wavelet analysis of signals with fractal analysis algorithms since the width of the multifractal spectrum and the correlation dimension of the BP and BFV signals are informative since they can correctly show the relationship between these signals having different scaling behavior.

## 5. Conclusions

Ultimately, the application of time-frequency transformations makes it possible to optimize the timing of obtaining the necessary information about the state of CA in order to study the mechanisms of the regulation of cerebral blood flow and make therapeutic and tactical decisions in real time, including in intensive care.

The results of our studies reflect the reliability of the data obtained from the non-invasive assessment of the CA rate in real-time and allow us to identify a group of patients with impaired CA and a high risk of complications and to determine the tactics and continuous monitoring of treatment results.

The comparative analyses of the Fourier and wavelet transform show certain advantages of the latter when carrying out standardized loads and allow us to recommend using the wavelet transform to build algorithms for processing indicators from systemic and cerebral hemodynamics.

## Figures and Tables

**Figure 1 biomedicines-10-03057-f001:**
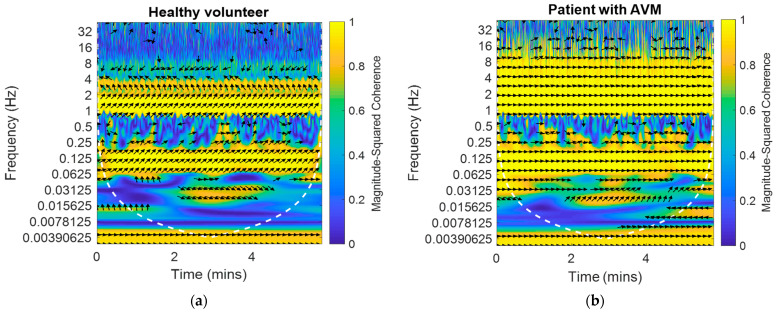
The example of the dependences of the phase shift between the BP and BFV signals in the left and right middle cerebral arteries against the background of the magnitude-squared coherence between the signals on time and pseudo-frequency: (**a**) For a healthy volunteer on one side; (**b**) For a patient with an arteriovenous malformation in the left middle cerebral artery on the AVM side.

**Figure 2 biomedicines-10-03057-f002:**
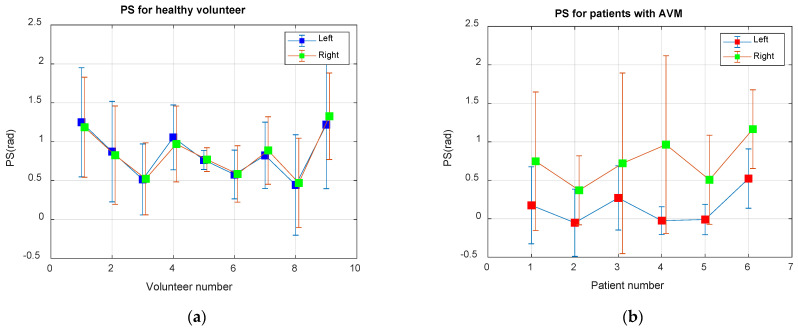
Average phase shift (PS) between the BP and BFV signals in the Mayer waveband: (**a**) For nine healthy volunteers; (**b**) For six patients with an AVM in the area of the left middle cerebral artery.

**Figure 3 biomedicines-10-03057-f003:**
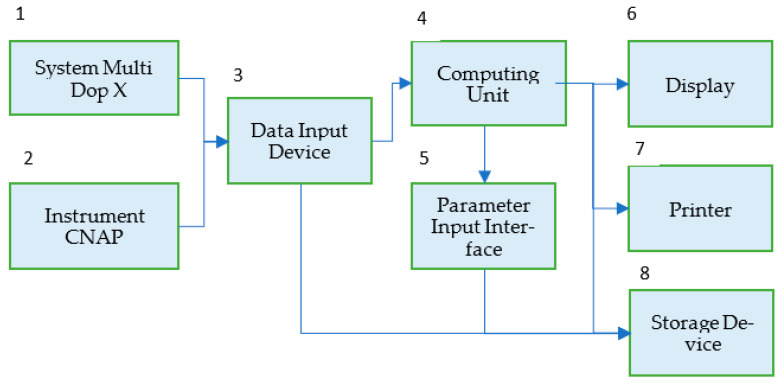
Block diagram of the NCM system for monitoring the state of the target audience in real time.

**Figure 4 biomedicines-10-03057-f004:**
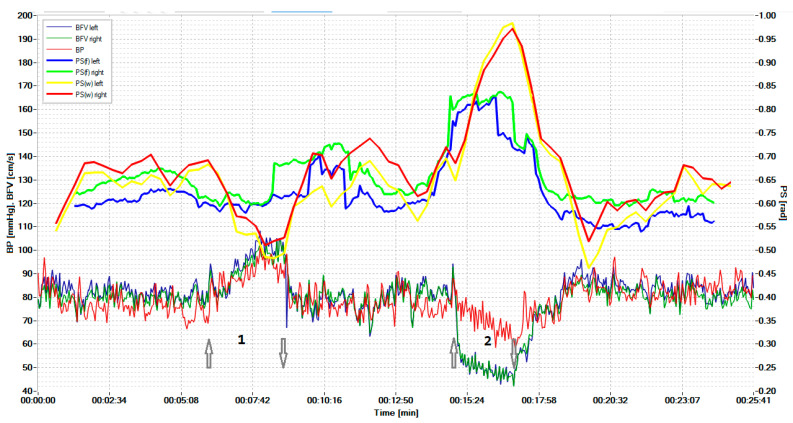
Simultaneous monitoring of parameters of the cerebral, systemic hemodynamics, and the cerebral autoregulation state in a 19-year-old healthy volunteer. The arrow pointing up indicates the beginning of the hypercapnic test (1), while the arrow pointing down indicates its end. Similarly directed arrows indicate the beginning and end of the hypocapnic test (2).

**Figure 5 biomedicines-10-03057-f005:**
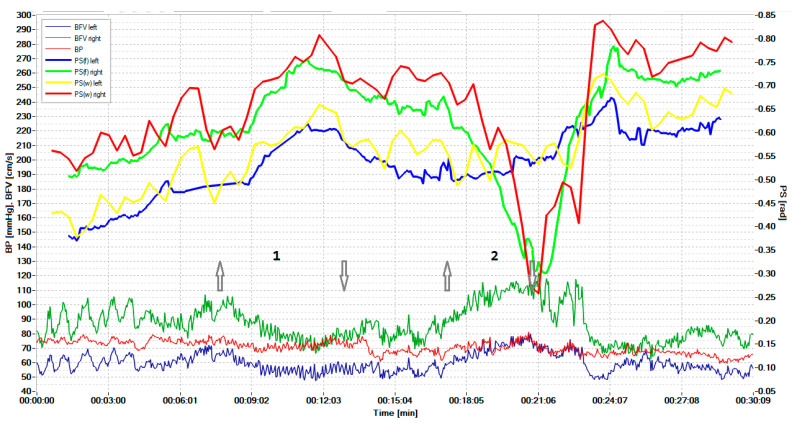
Simultaneous monitoring of the cerebral, systemic hemodynamics, and the state of the CA in a 73-year-old patient with critical stenosis of the internal carotid artery. The arrow pointing up indicates the beginning of the hypercapnic test (1), while the arrow pointing down indicates its end. Similarly directed arrows indicate the beginning and end of the hypocapnic test (2).

**Figure 6 biomedicines-10-03057-f006:**
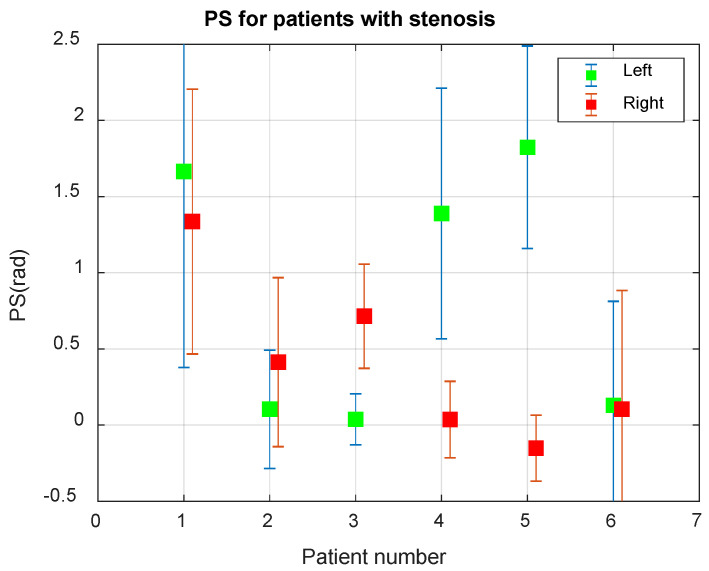
Average phase shift (PS) between the BP and BFV signals in the Mayer waveband in 6 patients with carotid stenosis.

## Data Availability

Not applicable here.

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
