# Peer review of "Investigation of Cerebral Autoregulation Using Time-Frequency Transformations"

_biomedicines, 2022, doi:10.3390/biomedicines10123057_

Round 1
Reviewer 1 Report
The authors of “Investigation of cerebral autoregulation using time-frequency transformations” describe how wavelet and Fourier transform based coherence analysis of Mayer waves compare between healthy and patient populations. The authors found that, according to the data shown in this article, Fourier and wavelet analysis provide comparable results. I enjoyed the potential of this work, the data available and the gist of the paper but have severe concerns about the focus of the presented text and it's English language quality. I have laid out my detailed criticism below, based on which I recommend a major revision of the text.
1) First, I like to recommend seeking support for the English language. I noticed grammatical errors, use of colloquial language, as well as repetitions of entire paragraphs throughout the document. Not everybody has Mastered the English language perfectly, so this is a major point but one that is easily addressed by seeking professional editing help. Here are just a few examples as there are too many to list:
a. Insufficiently good = poor (L. 75)
b. Multiple cases of too many or too few “and” and “the”
c. Absolute statements such as “Wavelet transforms have all the advantages of Fourier transforms” (L. 187-188) are overstating the truth and unnecessary as they do not add much information.
d. Frequency localization increases (L. 223) and temporal localization of the signal decreases (L. 224) – These statements are confusing. The authors likely meant that the temporal resolution is decreasing as the frequency resolution increases.
2) The general structure of the paper could be improved. The authors have placed important results in the discussion, such as the number of data points and the found range of Mayer waves, which should be listed or even graphed as a result put into context of current literature in the discussion. Statements such as “Median filtering instead of low-pass filtering of the wavelet coefficients in time and scale […] ” (L. 257f) are generally considered discussion points or future work, but do not belong inside the methods section as these methods were not used to generate the results shown.
3) General: I thing a comparison and discussion of the different frequency bands mentioned in the introduction (heart rate, respiration, Mayer, B-waves) and their ability to predict CA using Fourier VS Wavelet would make for a great addition to this paper.
4) General: I was under the impression that this paper was aiming to compare the performance of Fourier transformation and Wavelet with respect to the Mayer waves. This aspect is not pointed out explicitly, there are no metrics that would suggest this analysis and there is no reference or discussion in the discussion section that could help the reader understand this. As it stands, what is the novelty and motivation behind this work?
Now let me go over individual chapters
5) Introduction: The introduction starts with a rather large section of generic statements about the advent of signal processing and introduction of computers into medicine, which appears to be out of place for this article. In fact, I would highly recommend to simply remove the lines 28 to 47, as they do not add to the paper at all. Rather, state the medical need relevant to cerebral autoregulation and the medical implications of cerebral autoregulation here.
6) Introduction: The literature overview is limited to ultrasound based measurements of dynamic autoregulation and fails to paint the broader picture. Other modalities such as NIRS, MRI, DCS, etc. are not mentioned and all literature is at least 10 years of age. I recommend searching out recent articles that have used methods similar to the work proposed by the authors. Furthermore, the literature seems to be limited to a research group around Czosnyka et al., which is a good reference, but does not represent the entirety of the field.
7) Introduction: As I understand it, the authors uniqueness lays in the use of Mayer waves as their primary source of information, which might give them an advantage over other oscillatory signals in the hemodynamic signal of the brain. The reason to chose this frequency range is not motivated in the introduction. The authors simply show frequency ranges (L. 64-66) and from there on out only focus on Mayer waves. The authors need to explain their choice with reference and reason.
8) Methods: The methods section is overall very good, very detailed and explicit with respect to the algorithm itself. I would suggest removing some of the excessive methods to improve legibility. For example, the authors explain their Fourier transformation based coherence and phase analysis in great detail (L. 104 – 157) only to later point out (L. 360) that they have previously published this in another paper. I would recommend citing this paper early and explain the differences, adding a brief summary, but not repeating the explanation here.
9) Methods: 2.1 is lacking a lot of detail about the performed experiments, especially with respect to hyperventilation procedures (duration, how set up, how controlled, how recorded), probe placement, the use of a specific gas mixture in later experiments. Were those approved by the Ethics Committee as well. The language here needs to be improved to be more explicit and detailed.
10) Methods: Is the frame shift and the window shift the same? Why is it redefined in line 144? Please clarify.
11) Methods: Lines 145 to 157 are hard to follow in text. Maybe an equation based explanation would be beneficial here!
12) Methods: A similar recommendation for the wavelet transform sections. I recommend referencing a good article on wavelet transforms, or a book, to cut down on the detailed explanation of what a wavelet transform is (entire section 2.2.2 not needed) and focus on how the authors used it specifically in for this work, naming input parameters, window sizes, and chosen wavelet type.
13) Methods: Ranges in equations (e.g. Eq. 9) are provided from 0 to N-1, which is a programming annotation, not a mathematical annotation. This is especially confusing because the authors seem to use Matlab (not explicitly mentioned in the methods but it SHOULD BE), which indexes mathematically and not like other programming languages starting at 0.
14) Methods: The authors should provide more detail on how the data was collected, if the shown results were significant or not (statistical tests!), provide information on how the two methods were compared, etc. The focus point on the clinical implications is missing.
15) Results: Repetition of results is not needed at the beginning of the results.
16) Results: Figure should be capitalized when referencing a figure label.
17) Result: Shear angle (L. 291) is an expression not used before and commonly used in mechanical engineering. I assume the authors were refereeing to the phase difference between BP and BFV? Technical terms must be consistent throughout the text.
18) Result Figure 2: From personal experience I believe these to be the output plots of the matlab function “wcoherence”. If that is the case, the authors MUST cite it in the methods section and give credit to Mathworks Inc.
19) Result Figure 3: Add significance indication to the plot (and explain how calculated in the methods). More of these as a comparison between wavelet and FFT would help tell the story of this paper and make it significantly different from the previous publication of the same group.
20) Result Figure 4: This figure shows an overview of the real time processing structure, but unfortunately is does not provide a lot of information. It would be beneficial to add information such as window size, how M-wave ranges were chosen, the placement of probes on a subject, etc.
21) Result Figure 5: The use of hyperventilation and inspiratory gas mixtures were not explained in the methods! The axis labels and numbers are hard to read.
22) Discussion: Line 360 to 362 is a key statement and must be moved to the motivation part in the introduction.
23) Discussion: The authors mention that special metrics were calculated to proof the benefit of using Wavelet over discrete Fourier transformation (L. 362). However, concluding graphs, metrics and discussion of the results were not obviously conveying this message. I suggest showing these results and making this the major argumentation point of this work, in combination with the chosen frequency range. Comparative figures and statistics are missing.
24) Discussion: (L. 363-370) This is results!
25) Discussion: The discussion is not picking up on the literature provided in the beginning, making it hard to understand the significance of this work in the context of current literature.
26) References: I noticed that the references were cut off after 3 authors with the use of et al. Is this intentional? I believe there is no such rule in the author guidelines. Correct me if I am wrong, but please list all authors for all references.
Reviewer 2 Report
The work proposes the investigation of cerebral autoregulation by means of time frequency techniques exploiting and comparing two approaches, one based on the short time Fourier transform and the latter based on Wavelet functions. The methods were applied in healthy subjects and in patients with malformations or stenosis of the brain arteries.
The work is interesting and proposes a practical solution that could have a relevant clinical impact.
However, I have several concerns that must be addressed before the paper can be considered for publication.
- Introduction should make reference to other works as those trying to assess phase shift or delay between BFV and BP (e.g. (2021) Biomedical Signal Processing and Control, 68 , art. no. 102735) or other techniques to assess CA (e.g. Auton Neurosci. 2022 Jan;237:102920.; J Clin Monit Compu. 2022 Jan 24).
Furthermore, the scope of the work is not very clear and should be better stated.
- Patients: please provide age and gender of patients and healthy subjects, in terms of mean and SD. State whether the two groups are similar or not, and, if this is not the case, include this fact in the discussion when comparing the two populations (see also the comment related to statistical analysis).
- Were patients monolaterally or bilaterally impaired? Was there any difference in BFV signals in the two arteries?
- Did you perform mean BP and mean BFV time series extraction before assessing Fourier Transform and any other subsequent analysis? Authors should speak in terms of time series and not of original raw signals. E.g. in figure 5 and 6 what are the signals shown? According to their absolute values and waveforms/resolution they should not be raw BP and BFV signals but time series…
The resolution of the signals/time series must also be provided. The time series extraction (if any) should be described.
- Statistical analysis is entirely missing, if two methods (short time Fourier transform and Wavelet) are compared, a statistical analysis should be reported. Statistics should be performed to assess differences a. left vs right; b. patients vs healthy subjects…
- Line 273 and following: why author chose to have H2>0.7 as a prerequisite? In different points a different value is chosen (e.g. 0.7 line 273, 0.6 at line 284). This must be justified with appropriate citations and unified. See e.g.[ Physiol. Meas. 37 (2016) 661–672] for significance of coherence.
- Later, at line 384 it is not clear why authors speak in terms of H2<0.6. Was it necessary H2>0.7 or H2<0.6 and for what reason?
- Line 279: “with an interval of 0.01”: what does it mean? Was there a delay between signals? Did you synchronize them before performing phase shift computation?
- How was fig.2 obtained as related to left and right signals? Were they merged? Was there any difference between the two side? I would expect so in panel b, since it is referred to a patient with impairment in the left side.
- Was the different PS in left and right side for patients shown in Figure 3 due to a monolateral impairment?
- Figure 4: was the monitoring system already implemented? Do you have a picture of it?
Figure 5 and 6: if an hypercapnic and hypocapnic test were performed, it should be stated in the experimental protocol. The different experimental conditions must be compared with appropriate one way or two way statistical analysis.
- Statistical analysis should be described in an appropriate paragraph as said above.
Results must describe quantitative findings more than qualitative ones.
E.g. are there any significant differences between groups and conditions?
Quantify them with appropriate data (E.g. add tables with mean and SD of PS , H^2 and other standard indices of CA starting from mean and variance of mean CBV and BP. See e.g. white paper on CA, J. Cereb. Blood Flow Metab. 36 665–80).
- What are PS(f) and PS(w) shown in figs.5,6 referred to? Please explain everything in text and captions.
The colours of PS(f) right and PS(w) right are very similar, please modify.
Why in Fig.5 PS(f) left is very close to PS(w) right while in Fig.6 PS(f) left is close to PS(w) left? Is it correctly represented? Is there any physiological reasons under these facts?
- Line 365: it is not clear what authors mean. If this are results, numbers should be reported in results paragraph and only discussed in this section.
- Conclusions should be better supported by data.
Round 2
Reviewer 2 Report
After the first revision the paper has been improved but it is still not suitable for publication.
The healthy subjects appear significantly younger than the pathological groups. That could lead to differentiation in CA and other parameters not due to pathology but to age and should be discussed.
The statistical analysis explanation is still missing.
The choice of using H>0.6 should be better motivated.
